# Bioacetoin Production by *Bacillus subtilis* subsp. *subtilis* Using Enzymatic Hydrolysate of Lignocellulosic Biomass

Meenaxi Saini [1], Anu [1], Alexander Rapoport [2,*], Santosh Kumar Tiwari [3], Davender Singh [4], Vinay Malik [5], Sandeep Kumar [6] and Bijender Singh [1,7,*]

[1] Laboratory of Bioprocess Technology, Department of Microbiology, Maharshi Dayanand University, Rohtak 124001, HR, India; meenaxi1714@gmail.com (M.S.)

[2] Laboratory of Cell Biology, Institute of Microbiology and Biotechnology, University of Latvia, Jelgavas Str., 1-537, LV-1004 Riga, Latvia

[3] Department of Genetics, Maharshi Dayanand University, Rohtak 124001, HR, India; santoshgenetics@mdurohtak.ac.in

[4] Department of Physics, RPS Degree College, Balana, Mahendergarh 123029, HR, India

[5] Department of Zoology, Maharshi Dayanand University, Rohtak 124001, HR, India; vinaymalikzoo@mdurohtak.ac.in

[6] Department of Biotechnology, Shobhit Institute of Engineering and Technology, Modipurum, Meerut 250110, UP, India

[7] Department of Biotechnology, Central University of Haryana, Jant-Pali, Mahendergarh 123031, HR, India

*   Correspondence: rapoport@mail.eunet.lv (A.R.); ohlanbs@gmail.com (B.S.)

**Abstract:** Acetoin is an important bio-product useful in the chemical, food and pharmaceutical industries. Microbial fermentation is the major process for the production of bioacetoin, as the petroleum resources used in chemical methods are depleting day by day. Bioacetoin production using wild microorganisms is an easy, eco-friendly and economical method for the production of bioacetoin. In the present study, culture conditions and nutritional requirements were optimized for bioacetoin production by a wild and non-pathogenic strain of *B. subtilis* subsp. *subtilis* JJBS250. The bacterial culture produced maximum bioacetoin (259 mg L$^{-1}$) using peptone (3%) and sucrose (2%) at 30 °C, 150 rpm and pH 7.0 after 24 h. Further supplementation of combinatorial nitrogen sources, i.e., peptone (1%) and urea (0.5%), resulted in enhanced titre of bioacetoin (1017 mg L$^{-1}$) by the bacterial culture. An approximately 46.22–fold improvement in bioacetoin production was achieved after the optimization process. The analysis of samples using thin layer chromatography confirmed the presence of bioacetoin in the culture filtrate. The enzymatic hydrolysate was obtained by saccharification of pretreated rice straw and sugarcane bagasse using cellulase from *Myceliophthora thermophila*. Fermentation of the enzymatic hydrolysate (3%) of pretreated rice straw and sugarcane bagasse by the bacterial culture resulted in 210 and 473.17 mgL$^{-1}$ bioacetoin, respectively. Enzymatic hydrolysates supplemented with peptone as a nitrogen source showed a two to four-fold improvement in the production of bioacetoin. Results have demonstrated the utility of wild type *B. subtilis* subsp. *subtilis* JJBS250 as a potential source for economical bioacetoin production by making use of renewable and cost-effective lignocellulosic substrate. Therefore, this study will help in the sustainable management of agricultural waste for the industrial production of bioacetoin, and in combating environmental pollution.

**Keywords:** *Bacillus subtilis* subsp. *subtilis* JJBS250; bioacetoin; optimization; lignocellulosic biomass; enzymatic hydrolysate

## 1. Introduction

Bioacetoin is naturally present in various plants, fermented foods and fruits. Bioacetoin is usually considered as a fragrance agent and flavor enhancer in the cosmetic and food industries [1–3]. Bioacetoin also acts as a promising building-block for the production of

various liquid hydrocarbon fuels [4,5], and therefore, is widely used in chemical industries, and also acts as a bioactive molecule, so is widely preferred in agricultural applications. The products obtained from bioacetoin can be used in the different industries related to pharmaceutical, information technology, paint and chemical synthesis. The microbial fermented products, like vinegar and alcoholic beverages, are rich sources of bioacetoin. Bioacetoin and its derivatives are often detected during the component analysis of different foods through mass spectrometry, gas chromatography and other techniques [6]. Bioacetoin is usually produced via chemical synthesis using petroleum-related substrates as raw materials, which are known to cause adverse effects on the environment [2,7,8]. The production cost of acetoin is highly affected due to the depletion of fossil fuels, as well as by the fluctuation in the price of crude oil. Furthermore, the chemical methods produce toxic byproducts, thereby limiting its usage in the medicine, food and pharmaceutical industries [2,7]. In view of this, an eco-friendly approach, i.e., microbial fermentation has been proven as an attractive alternative. Bioacetoin can be produced by numerous microorganisms, including *Bacillus subtilis*, *B. amyloliquefaciens*, *Paenibacillus polymyxa*, *Enterobacter cloacae*, *Serratia marcescens* and *Saccharomyces cerevisiae* [2,3,5,6,8,9]. The production of bioacetoin by bacteria occurs via two steps; the first step involves acetolactate synthase, which produces one $CO_2$ molecule, and the second step involves acetolactate decarboxylase that leads to bioacetoin formation by releasing a second $CO_2$ molecule [9]. *Bacillus subtilis* is a promising strain for bioacetoin production, having characteristics of high sugar consumption. The strains of *Bacillus subtilis* can grow efficiently on cost-effective carbon and nitrogen sources for production of various acetoin and other products. Microbial fermentation mainly uses glucose and other sugars for the production of bioacetoin. This makes the process costly for the production of bioacetoin at the industrial level [7]. However, microbial fermentation is especially interesting, when based on the use of cost-effective and easily available renewable residues as the substrates. This will result in a reduction of the overall production cost of the bioacetoin and other value-added products/metabolites [4,7].

Bioprocess development is an important parameter for the cultivation of microorganisms for the industrial production of useful products/metabolites. The selection of suitable and optimal culture conditions is the first step of any bioprocess based on selected microorganisms [5,10]. Microbial growth and metabolism are significantly affected by both physical and chemical factors. The optimal level of each factor is necessary for high metabolite/product formation during microbial fermentation. Therefore, the optimization of physical and chemical factors is an important step in the development of a bioprocess [4,11,12]. The one variable at a time (OVAT) approach is an important preliminary strategy for the optimization of culture conditions in submerged fermentation. In this approach, one factor is varied at one time, while other factors are kept constant. The initial step in any microorganism-based bioprocess is to select suitable and optimal culture conditions [5,6]. Achieving an optimal level of each factor is essential for maximizing the production of metabolites and products during microbial fermentation. Thus, it is important to optimize these physical and chemical factors to develop an effective and economical bioprocess for the production of bioacetoin [4,11,12].

Lignocellulosic biomass is the sustainable and renewable resource for the production of value-added products using microorganisms. Lignocellulosic biomass is generated in huge quantities after crop harvesting [2,8,11–13]. There is no suitable method available for the proper disposal and management of this biomass; therefore, farmers are forced to burn this biomass in open fields. This practice releases toxic pollutants in the environment and causes health problems in human beings. Therefore, the utilization of lignocellulosic biomass as a substrate for the cultivation of microorganisms is preferred for the production of bioacetoin at the industrial level [8,11,14]. Lignocellulosic biomass is a complex structure composed of cellulose, hemicellulose and lignin, which are hydrolysed into monomers for the production of valuable products, including biofuels, biopolymers, biopesticides, etc., using microorganisms [2,4,9,13,15–17]. The presence of lignin is a major barrier for the efficient bioconversion of cellulose and hemicellulose into monomeric sugars. The pretreat-

ment of lignocellulosic biomass is a useful strategy for the removal of lignin and increasing the availability of cellulose and hemicellulose for enzymatic hydrolysis [11,12,16,17]. Various physical, chemical and biological methods have been employed for the pretreatment of lignocellulosic biomass. Physico-chemical methods of pretreatment have been preferred due to high lignin removal and improved enzymatic hydrolysis. Usually, the biomass is pretreated to disrupt its complex and recalcitrant nature, thus resulting in the release of carbohydrates or sugars that are utilized by microorganisms for the production of bioacetoin [13,14,16–18]. But there is a need for ample supply of the respective biomass with simple processing for effective utilization at the industrial scale. Both rice straw and sugarcane bagasse are luxuriantly available plant biomasses for utilization as a substrate for microbial fermentation in order to produce bioacetoin by microorganisms. These plant biomasses are pretreated with a physio-chemical method followed by saccharification using microbial cellulases. The resultant enzymatic hydrolysates containing sugars can serve as a suitable medium for the microbial fermentation and economical production of bioacetoin [4,11].

There is scanty information on bioacetoin production by the wild and non-pathogenic microorganisms, as the majority of the available reports have involved the use of genetically modified microorganisms [2]. Strain improvement using genetic engineering is a cumbersome and costly process for the generation of improved and modified microorganisms. Bioacetoin produced by pathogenic microorganisms cannot be utilized for application in food and medicines. Therefore, the use of wild and non-pathogenic microbial sources is the best alternative for the production of bioacetoin suitable for the food, medicine and pharmaceutical industries [2,4,7]. Therefore, we have isolated a wild strain of *B. subtilis* subsp. *subtilis* JJBS250 from soil samples, and selected it as a potent producer of bioacetoin in submerged fermentation. The present study reports the possibility of the improved production of bioacetoin by a wild strain of *B. subtilis* subsp. *subtilis* JJBS250 under optimized culture conditions in submerged fermentation. Furthermore, the enzymatic hydrolysate of pretreated lignocellulosic biomass also supported bioacetoin production that can be used in the food, medicine and pharmaceutical industries. Therefore, the present study reports on the economical acetoin production by employing waste or residues of lignocellulosic substrates, as well as the mitigation of environmental pollution due to the sustainable management of solid waste.

## 2. Materials and Methods

### 2.1. Source of Bacterial Strain

The bacterial culture *Bacillus subtilis* subsp. *subtilis* JJBS250 was isolated from soil samples using a serial dilution method as described by Jain and Singh [19]. The bacterial culture was identified on the basis of morphological features, biochemical tests and 16S rDNA sequencing [19]. The culture was maintained on nutrient agar plates at 4 °C, as well as glycerol stocks at −20 °C. The bacterial culture was grown in nutrient broth medium at 30 °C and 200 rpm for 24 h. Metabolically active growing culture (24 h) was used as inoculum for bioacetoin production.

### 2.2. Production of Bioacetoin by Bacillus subtilis subsp. subtilis JJBS250

The bacterial culture was grown in Erlenmeyer flasks (250 mL) containing medium comprising ($w/v\%$) glucose 2, peptone 1 and $K_2HPO_4$ 0.1 (pH 7.0). The medium was autoclaved at 110 °C and 10 psi for 15 min. After cooling at room temperature, medium was inoculated with overnight grown the bacterial culture (2% $v/v$) and incubated at 30 °C and 150 rpm for 24 h. The bacterial culture was centrifuged at 10,000 rpm and 4 °C for 10 min. Culture filtrate was used for the estimation of bioacetoin.

### 2.3. Estimation of Bioacetoin

Bioacetoin was measured in the culture supernatant using the modified method described earlier [20]. Briefly, culture supernatant (1 mL) was mixed with 140 µL creatine

(0.5%) and mixed thoroughly. Reagent-A (500 µL of 5% 1-nepthol) was added to the mixture followed by the addition of reagent B (500 µL of 20% KOH), and the reaction mixture was mixed properly using vortex mixture and incubated at room temperature for 20 min. The absorbance of the mixture was recorded at 518 nm. The concentration of bioacetoin was estimated from the standard curve prepared from commercial bioacetoin using the same procedure.

### 2.4. Optimization of the Culture Conditions for Bioacetoin Production in Submerged Fermentation

Optimization of various factors was carried out in order to enhance production of bioacetoin by the bacterial culture in submerged fermentation. The effect of different incubation times (24–96 h), nitrogen sources (peptone, yeast extract, beef extract, urea and ammonium sulfate at 0.5%), sucrose concentrations (1–5%) and combinations of nitrogen sources (peptone 1% and urea 0.5%, yeast extract 1% and urea 0.5%, peptone 1%, yeast extract 1% and urea 0.5%) was studied on the production of bioacetoin by *Bacillus subtilis* subsp. *subtilis* JJBS250. Samples were taken after desired time intervals. The samples were centrifuged at 10,000 rpm and 4 °C for 10 min, and the cell-free culture supernatant was used for the estimation of bioacetoin.

### 2.5. Analysis of the Presence of Bioacetoin by Thin Layer Chromatography

Thin layer chromatography (TLC) was used to determine the presence of bioacetoin in the culture filtrate. The samples were spotted and allowed to run in running buffer containing *n*-butanol: ethanol: water in a ratio of 5:3:2. After the run, the plates were incubated with resublimed iodine for visualizing the bioacetoin.

### 2.6. Physico-Chemical Pretreatment of Lignocellulosic Biomass for Bioacetoin Production

Rice straw and sugarcane bagasse were collected locally and utilized in the experiments. Substrates were ground in a mixer grinder to get a size of 3 to 5 mm. Pretreatment of rice straw (10% *w/v*) was carried out with sodium carbonate [1% (*w/v*) at 121 °C, 15 psi for 60 min] as described earlier [21]. Pretreated biomass was washed with tap water followed by distilled water, and stored in air-tight polythene bags after drying at 50 °C till further use for enzymatic hydrolysis. Sugarcane bagasse (10% *w/v*) was steam pretreated at 121 °C and 15 psi for 60 min. Pretreated bagasse was dried and stored in air-tight polythene bags till further use for enzymatic hydrolysis.

### 2.7. Cellulase Production and Enzymatic Saccharification of Pretreated Lignocellulosic Biomass for Bioacetoin Production

Pretreated substrates were enzymatically hydrolysed using cellulase of a thermophilic mould *Myceliophthora thermophila* BJTLRMDU3. The mould was cultivated in solid state fermentation using 5 g rice straw (in 250 mL flasks) at a substrate to moisture ratio of 1:4 (pH 5.0), followed by autoclaving at 121 °C for 20 min, and incubated at 45 °C for 4 days. Cellulolytic enzymes were extracted using distilled water (20 mL/g) containing Tween 80 (0.1% *v/v*) at shaking conditions (200 rpm) for 1 h. The filtered clear culture supernatant was used for cellulase activity as described earlier [13,22,23]. Cellulase assay was carried out using carboxymethyl cellulose (0.5% *w/v*) as a substrate at 60 °C and pH 5.0 (0.1 M sodium acetate buffer), as described earlier [13,22,24]. Reducing sugars were estimated using the dinitro salicyclic acid method as described earlier [21]. One cellulase unit (IU) is the amount of cellulase required to liberate 1 µmole of glucose per min under the assay conditions. The pretreated rice straw and sugarcane bagasse were saccharified using crude cellulase (20 U/g) of *Myceliophthora thermophila* at 60 °C and pH 5.0 for 24 h [21,25].

### 2.8. Fermentation of Enzymatic Hydrolysate of Rice Straw and Sugarcane Bagasse for Bioacetoin Production by Bacterial Culture

Enzymatic hydrolysates were concentrated using lyophilisation, and used for the production of bioacetoin. Fermentation of enzymatic hydrolysates (equivalent to 3% sugars) was carried out for bioacetoin production by the bacterial culture with nitrogen source

(peptone) or without any nitrogen source in 250 mL Erlenmeyer flasks at shaking conditions in an incubator at 30 °C and 150 rpm for 96 h. The samples were taken and analyzed for bioacetoin production.

### 2.9. Statistical Analysis

All experiments were performed in triplicates, and their average values with standard deviation are presented. Statistical analyses were performed using GraphPad Prism software version 9.0 (GraphPad Software, San Diego, CA, USA). Significant differences ($p < 0.05$) were analyzed using one-way ANOVA and Tukey's post hoc tests. Graphs representing the correlations between variables were obtained by Sigma plot (version 10.0).

## 3. Results and Discussion

### 3.1. Optimization of Bioacetoin Production by Bacterial Culture

Acetoin is used in the food, pharmaceutical and chemical industries for the production of various products [2]. It has been listed as one of thirty bio-based platform products by the US Department of Energy, and therefore, more emphasis has been placed on the economical production of bioacetoin [2,5,6,10]. The composition of medium and culture conditions plays an important role in enhancing the production of a specific target compound by a given strain during the fermentation process [6,10]. In the present study, the optimization of culture conditions for bioacetoin production by *B. subtilis* subsp. *subtilis* JJBS250 was carried out. Incubation time is a critical factor, and results indicated that *B. subtilis* subsp. *subtilis* JJBS250 produced maximum bioacetoin (22 mg L$^{-1}$) after 24 h under shaking conditions (30 °C, 150 rpm), followed by a decline afterwards (Figure 1). Incubation time is important factor affecting microbial growth and metabolite production. The wild strain of *B. subtilis* subsp. *subtilis* JJBS250 produced bioacetoin maximally during the log phase. The decline in the production of bioacetoin after 24 h could be either due to the accumulation of toxic components in the culture medium or the depletion of nutrients [26]. Under these conditions, the microbe tends to undergo dormancy and stops its metabolic activities, hence, reducing the bioacetoin production [26]. Various studies have clearly observed the production of bioacetoin by microorganisms during the log phase and early stationary phase. Tian et al. [27] reported high bioacetoin production by *Bacillus subtilis* SF4–3 after 72 h. Mutant thermotolerant strain *Bacillus subtilis* IPE5–4-UD-4 also produced higher bioacetoin after 72 h under shaking conditions [28]. *Bacillus subtilis* BS7432 and BS7431 produced maximum bioacetoin after 60 and 70 h, respectively [29]. A genetically engineered strain of *Saccharomyces cerevisiae* JHY617-SDN produced higher bioacetoin in fed-batch fermentation in YEPD medium after 55 h as compared to batch flask fermentation after 48 h [3]. Microorganisms have been reported for maximum bioacetoin production in the range of 24 to 96 h post incubation in submerged fermentation.

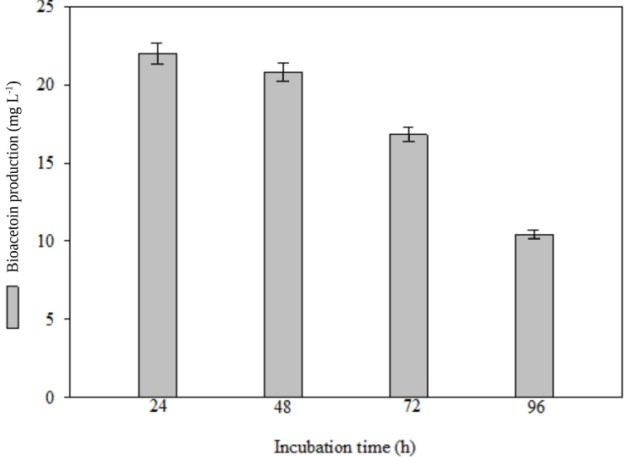

**Figure 1.** Effect of incubation time on bioacetoin production by *B. subtilis* subsp. *subtilis* JJBS250 at 30 °C and 150 rpm.

A suitable nitrogen source is needed for the optimal production of metabolites by the microorganisms (26). Complex organic nitrogen sources play an important role in the production of bioacetoin by microorganisms. Among different nitrogen sources, maximum bioacetoin production (75 mg L$^{-1}$) was attained in peptone-containing medium as compared to others (Figure 2a). Further, among different concentrations of peptone, supplementation of 3% peptone enhanced the bioacetoin (216 mg L$^{-1}$) production significantly (Figure 2b). There was a 3.0-fold improvement in bioacetoin production after the addition of 3% peptone in the medium. Organic nitrogen sources have been preferred for high bioacetoin production by the microorganisms [20,27]. Organic sources of nitrogen provide not only nitrogen, but also serve as sources of vitamins, minerals and other nutrients for the growth and metabolism of microorganisms [20,27]. Therefore, the presence of optimal levels of nitrogen sources is needed for higher product yield. *Bacillus subtilis* SF4–3 produced maximum bioacetoin using yeast extract after 72 h [27]. Taiwo et al. [20] reported high bioacetoin production by utilizing corn steep liquor as a nitrogen source by *B. subtilis* CICC10025 after 60 h. Xiao et al. [30] also reported high levels of bioacetoin using corn steep liquor as an organic nitrogen source by *Geobacillus* sp. XT15 at 55 °C. Similarly, Dai et al. [8] observed 35.5 g L$^{-1}$ bioacetoin using corn steep liquor powder as a nitrogen source by *Bacillus subtilis* CGMCC 13141. Furthermore, nitrogen sources supporting good microbial growth are responsible for better yields of metabolites [26].

Among different carbon sources, sucrose showed the maximum bioacetoin production (210 mg L$^{-1}$) by *B. subtilis* subsp. *subtilis* JJBS250 at 30 °C and pH 7.0 after 24 h (Figure 3a). Furthermore, among different concentrations of sucrose used, supplementation of 2% sucrose enhanced the production of bioacetoin by the bacterial culture, i.e., 259 mg L$^{-1}$ (Figure 3b). Bioacetoin is an intermediate compound of microbial overflow metabolism when the bacteria are cultivated in an environment containing an excessive glucose or other fermentable sugars [30]. Glucose has been the best carbon source for the production of bioacetoin by microorganisms due to its easy assimilation and metabolism. However, it has been reported that sucrose assimilation generates more energy than glucose for transport and metabolite synthesis; therefore, sucrose supported higher bioacetoin production by the bacterial culture. Tian et al. [27] reported enhanced bioacetoin production by *B. subtilis* SF4–3 after supplementation of glucose at pH 7.0. *Bacillus subtilis* ATCC 14884 produced 0.50 g L$^{-1}$ bioacetoin with 1.02 g L$^{-1}$ glucose in submerged fermentation [31]. Sharma and Noronha [16] observed that *Bacillus subtilis* 168 produced the maximum bioacetoin (0.08 g L$^{-1}$) in glucose-containing medium. Sharifi et al. [32] reported high levels of bioacetoin by *B. subtilis* GB03 after supplementation of glucose (75.91 g L$^{-1}$) as a carbon source in the medium. Recombinant *B. subtilis* ZB02 produced high bioacetoin by utilizing a combination of glucose, xylose and arabinose in the medium [33]. Similarly, recombinant *B. subtilis* F126 produced 22.04 g L$^{-1}$ biocetoin using glycerol as a carbon source [34].

The effect of the combination of different nitrogen sources on bioacetoin production was also investigated in this study, and it was found that the combination of peptone and urea significantly enhanced the bioacetoin production (1017 mg L$^{-1}$) by the bacterial culture (Figure 4). The combination of urea and peptone resulted in about a 4.0-fold increase in bioacetoin yield by *B. subtilis* subsp. *subtilis* JJBS250. This might be due to the fact that organic nitrogen sources vary in their nutrient composition [26]. The maximum bioacetoin production was achieved by the combination of yeast extract and corn steep dry by *Bacillus subtilis* SF4–3 [27], whereas *B. pumilus* ATCC 14884 produced maximum bioacetoin after supplementation of tryptone and yeast extract as nitrogen sources [31]. The genetically engineered strain of *Saccharomyces cerevisiae* JHY617-SDN produced maximum bioacetoin in fed-batch fermentation in YPD medium containing yeast extract and peptone [3]. Similarly, Xiao et al. [35] reported the production of 20.9 g L$^{-1}$ bioacetoin in medium containing yeast extract and peptone.

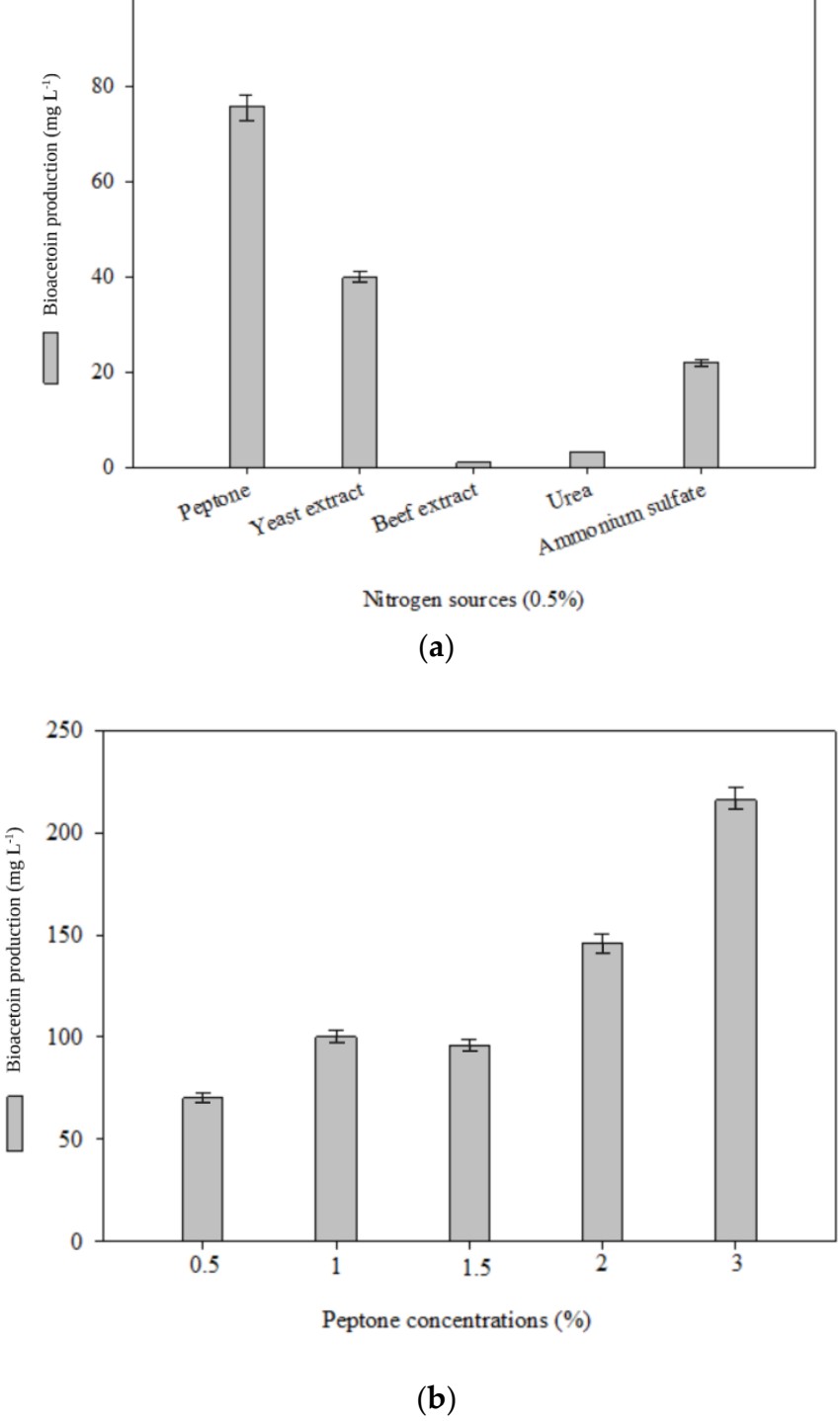

**Figure 2.** (**a**) Effect of different nitrogen sources (0.5%) on bioacetoin production by *B. subtilis* subsp. *subtilis* JJBS250 at 30 °C and 150 rpm after 24 h. (**b**) Effect of different concentrations of peptone (0.5–3%) on bioacetoin production by B. subtilis subsp. subtilis JJBS250 at 30 °C after 24 h.

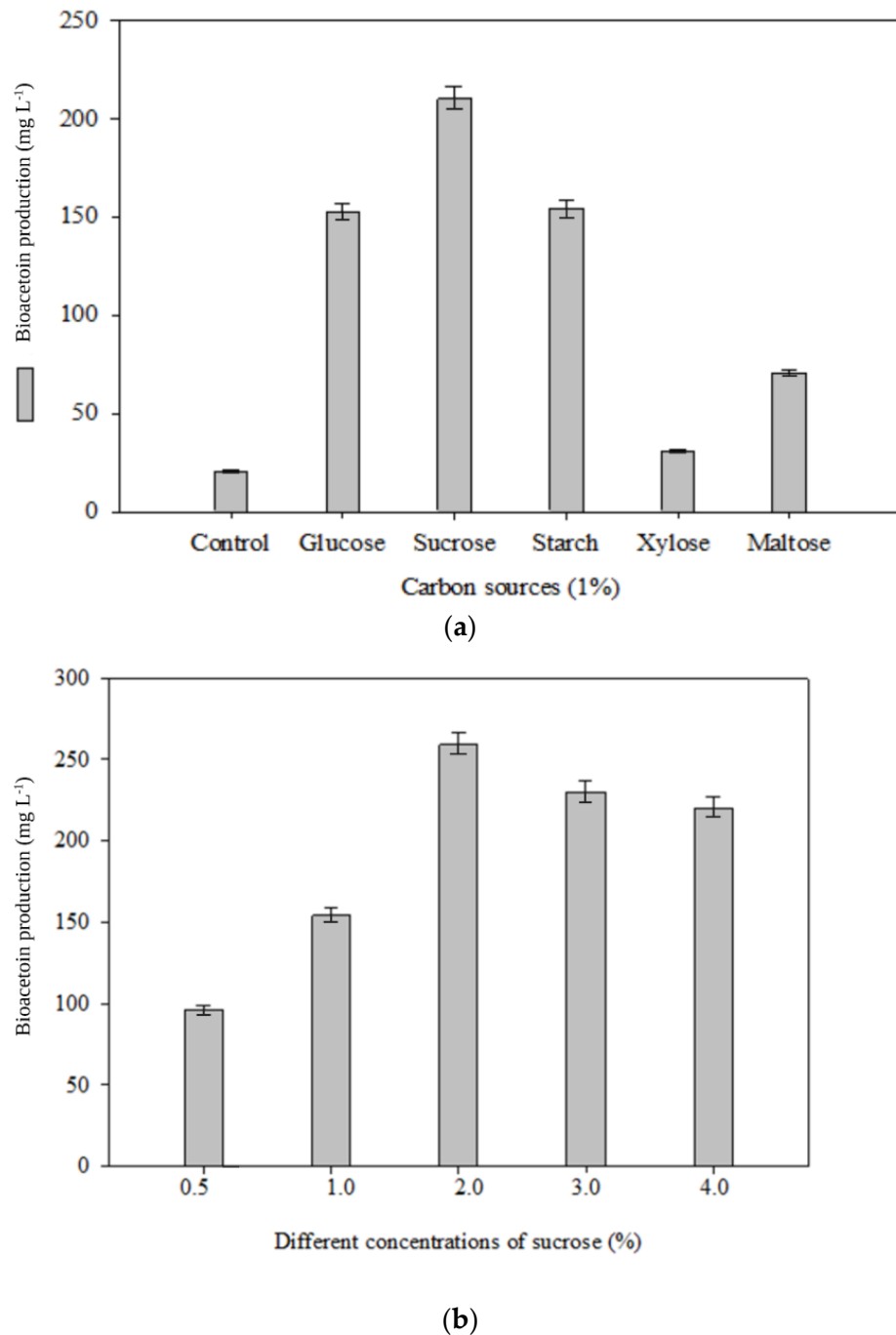

**Figure 3.** (**a**) Effect of different carbon sources (1%) on bioacetoin production by *B. subtilis* subsp. *subtilis* JJBS250 at 30 °C after 24 h. (**b**) Effect of different concentrations of sucrose (1–5%) on bioacetoin production by *B. subtilis* subsp. *subtilis* JJBS250 at 30 °C after 24 h.

*3.2. Analysis of Bioacetoin by Thin Layer Chromatography*

Thin layer chromatography analysis showed the presence of bioacetoin with an $R_f$ value of 0.31 cm, as measured with the standard bioacetoin sample (Figure 5). Bioacetoin showed yellow spots in the presence of iodine vapour. There is no report showing the analysis of bioacetoin using TLC in the literature. Thin layer chromatography is an easy and quick method for the qualitative and quantitative analysis of microbial metabolites [25]. This method has been used for the analysis of various microbial metabolites like amino acids, lipids, pigments, etc., which are of industrial importance [25].

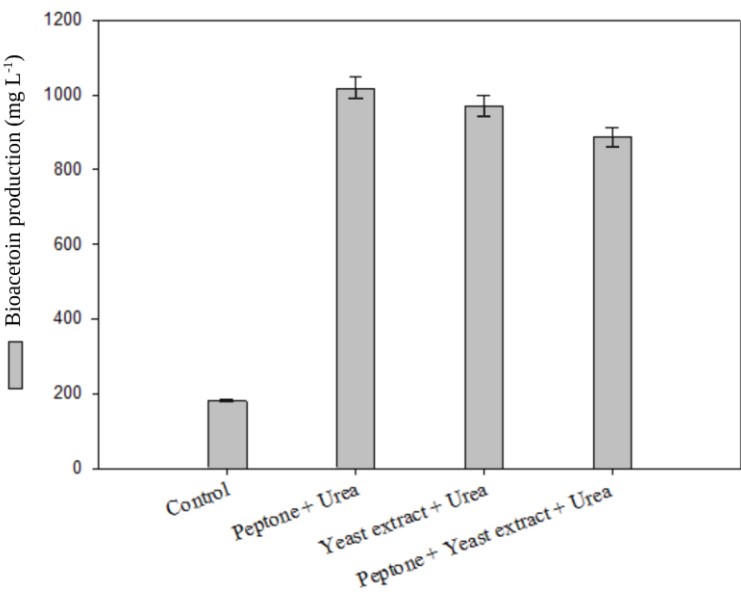

**Figure 4.** Effect of different combinations of nitrogen sources on bioacetoin production by *B. subtilis* subsp. *subtilis* JJBS250 at 30 °C after 24 h.

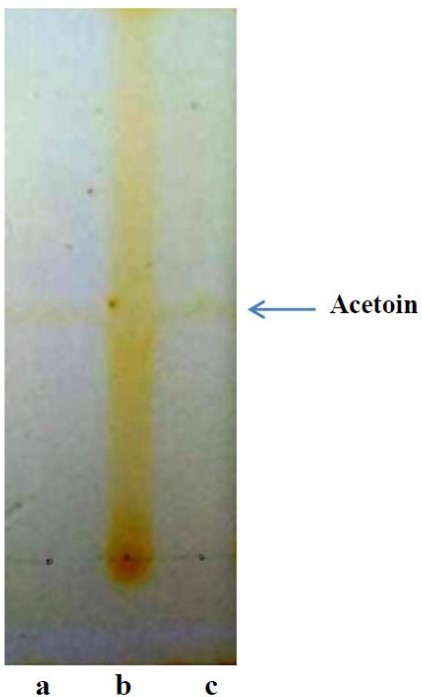

**Figure 5.** Analysis of bioacetoin using thin layer chromatography in bacterial culture filtrate, (**a**) Standard bioacetoin, (**b**) culture medium (0 h), (**c**) culture filtrate (24 h). Arrow indicates the position of bioacetoin.

### 3.3. Fermentation of Enzymatic Hydrolysate of Lignocellulosic Biomass for Bioacetoin Production

The fermentation of the enzymatic hydrolysate of rice straw without peptone by the bacterial culture produced maximum bioacetoin after 24 h, which is 4-fold higher (210 mg L$^{-1}$) compared to the peptone-containing the enzymatic hydrolysate of rice straw (53 mg L$^{-1}$) (Figure 6a). Microorganisms utilize sugars as the primary carbon source for bioacetoin production, making the process costly and unsuitable for industries [2,3,5,6,10]. Therefore, the use of lignocellulosic materials including rice straw, sugarcane bagasse, etc.,

could be a sustainable and economic alternative for the production of bioacetoin. The enzymatic hydrolysis of lignocellulosic biomass generates sugars, which can be fermented to bioacetoin using microorganisms [16,17]. Furthermore, the supplementation of peptone in the enzymatic hydrolysate of rice straw decreased the production of bioacetoin in the enzymatic hydrolysate, showing the suitability of the enzymatic hydrolysate alone as a medium for higher bioacetoin production. It has further reduced the cost of the medium for bioacetoin production significantly at the industrial level. As a possible explanation, the pretreatment and enzymatic hydrolysis might have resulted in the production of nitrogenous compounds from lignocellulosic substrates supporting the production of bioacetoin by the bacterial culture.

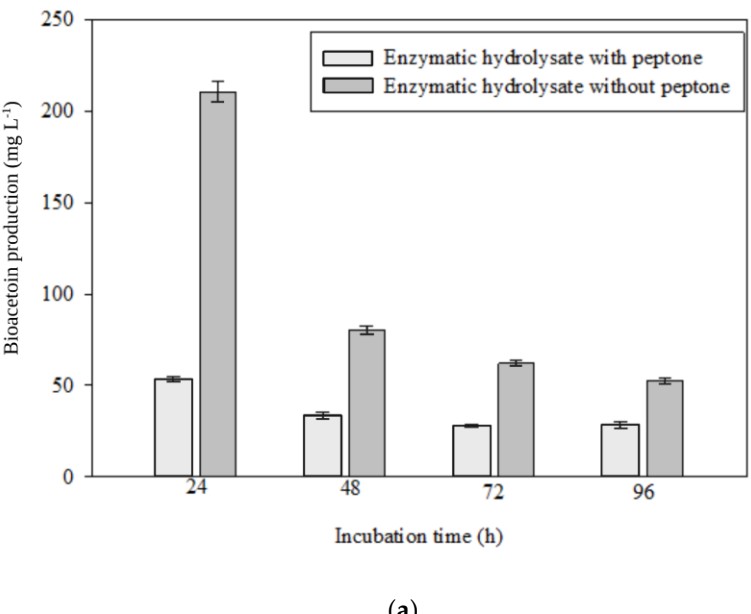

(**a**)

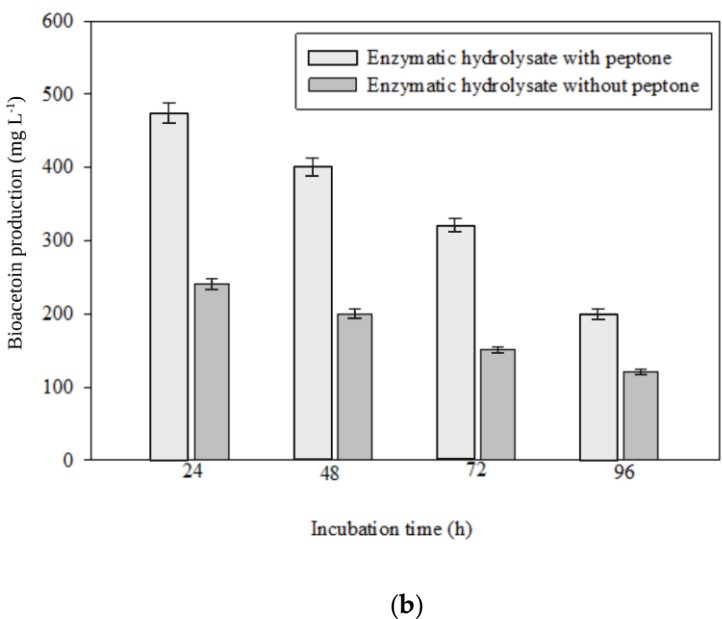

(**b**)

**Figure 6.** (**a**) Fermentation of rice straw hydrolysate by *B. subtilis* subsp. *subtilis* JJBS250 for bioacetoin production at different incubation times. (**b**) Fermentation of sugarcane bagasse hydrolysate by *B. subtilis* subsp. *subtilis* JJBS250 for bioacetoin production at different incubation times.

In contrast, the enzymatic hydrolysate of sugarcane bagasse supplemented with peptone resulted in higher bioacetoin production (473.17 mg $L^{-1}$) after 24 h, which is about two-fold higher than the medium lacking peptone (Figure 6b). The differential behavior of the enzymatic hydrolysates of two different lignocellulosic substrates is due to differences in their biomass composition and sugars in the hydrolysate as a result of saccharification by the cellulolytic enzymes of *M. thermophila*. Zhang et al. [33] reported high bioacetoin production by *Enterobacter cloacae* SDM 53 in pretreated corn stover hydrolysate. Xiao et al. [35] also reported high bioacetoin production in soya bean meal hydrolysate supplemented with molasses. Jia et al. [28] observed maximum bioacetoin production (22.76 g $L^{-1}$) from corncob hydrolysate by *B. subtilis* IPE5–4-UD-4 at 50 °C after 60 h. Recombinant *Bacillus subtilis* ZB02 produced 11.2 g $L^{-1}$ bioacetoin by utilizing corn stover and corn powder hydrolysate as a substrate [33]. Supplementation of yeast extract in the hydrolysate of oil palm mesocarp fiber enhanced the bioacetoin production by *E. coli* [36]. Similarly, supplementation of glucose in soybean meal hydrolysate resulted in high bioacetoin production by *Bacillus* sp. H-18W [37]. A comparative study of bioacetoin production by various microorganisms using lignocellulosic biomass is given in Table 1. From the data, it is clear that the majority of these studies have reported higher production of bioacetoin as compared to our study. This is due to the fact that these studies used genetically modified microorganisms for bioacetoin production and the present study showed enhanced bioacetoin production using the wild strain of *B. subtilis* subsp. *subtilis* JJBS250. Lignocellulosic biomass is generated in huge quantities after crop harvesting in India and other countries in the world [11,21,28]. Large amounts of lignocellulosic biomass are not managed properly, which leads to environmental and health problems. Plant biomass is burnt in open fields for its quick disposal and management. Therefore, the utilization of lignocellulosic biomass as a substrate for the production of bioacetoin could be an efficient method for sustainable management all over the world. The enzymatic hydrolysate of lignocellulosic biomass is a rich source of nutrients for the cultivation of microorganisms for bioacetoin production at the industrial level [2,21,28,33]. Therefore, the present study reports the development of a cost-efficient and eco-friendly bioprocess for the production of bioacetoin using the enzymatic hydrolysate of lignocellulosic biomass by a wild and non-pathogenic bacterial strain. However, bioacetoin production can further be enhanced by statistical designs and/or a fed-batch process using the enzymatic hydrolysate of either rice straw or sugarcane bagasse. The utilization of rice straw and sugarcane bagasse will be feasible and sustainable for scaling up of the bioprocess at the industrial level due to the availability of huge quantities of these lignocellulosic substrates.

**Table 1.** Comparison of culture conditions optimized for enhanced bioacetoin production by various microorganisms using lignocellulosic biomass.

| Microorganisms Involved | Type of Microorganism | Biomass Used | Fermentation Conditions | Acetoin Production (g $L^{-1}$) | References |
|---|---|---|---|---|---|
| *Bacillus subtilis* 168 | Recombinant | Pretreated okara | 37 °C, 36 h | 11.79 | [1] |
| *Bacillus subtilis* IPE5–4-UD-4 | Mutant | Pretreated corncob | 50 °C, 60 h | 22.76 | [28] |
| *Escherichia coli* | Recombinant | Pretreated oil palm mesocarp fiber | 37 °C, 24 h, 1000 rpm | 15.5 | [36] |
| *Zymomonas mobilis* 22C–BC5 | Recombinant | Pretreated corn stover | 33 °C, 24 h, 200 rpm | 10.0 | [38] |
| *Bacillus subtilis* ZB02 | Recombinant | Corn stover and corn powder hydrolysate | 37 °C, 30 h | 11.2 | [33] |
| *E. coli* DSM02-B | Recombinant | Pretreated brown seaweed | 37 °C, 96 h, 200 rpm | 4.80 | [39] |
| *Bacillus subtilis* 168 | Recombinant | Glucose | pH 7.5, 100 rpm, 37 °C, 36 h | 0.08 | [29] |
| *B. subtilis* subsp. *subtilis* JJBS250 | Wild | Pretreated rice straw Pretreated sugarcane bagasse | 30 °C, 24 h | 0.21 0.47 | Present study |

## 4. Conclusions

Optimization of culture conditions is an important step in submerged fermentation for the production of microbial products/metabolites. In the present study, the optimization of culture conditions resulted in a 46.22-fold enhancement in bioacetoin production by wild *B. subtilis* subsp. *subtilis* JJBS250. The enzymatic hydrolysate of rice straw supported maximum bioacetoin production without the addition of any nitrogen source, whereas sugarcane bagasse supported high bioacetoin production with supplementation of peptone as a nitrogen source. The results indicated the utility of low cost and luxuriantly available lignocellulosic residues as a convenient substrate for economical bioacetoin production by the bacterial culture. The utilization of rice straw and sugarcane bagasse may be a feasible, economical, eco-friendly and sustainable approach for scaling up the bioprocess at the industrial level in countries like India and Brazil, which are good producers of these lignocellulosic substrates during crop harvesting. Furthermore, a fed-batch fermentation strategy could be used for improving the yield of bioacetoin production by the *B. subtilis* subsp. *subtilis* JJBS250.

**Author Contributions:** Conceptualization, M.S., A., B.S. and A.R.; methodology, M.S., A., B.S., S.K.T. and A.R.; software, B.S., S.K.T., V.M. and S.K.; validation, M.S., A., B.S., S.K.T. and A.R.; formal analysis, M.S., A., B.S., V.M., S.K., D.S. and A.R.; investigation, M.S., A. and B.S.; resources, B.S., S.K.T., V.M. and D.S.; data curation, M.S., A., V.M., S.K.T., S.K. and B.S.; writing—original draft preparation, M.S., A. and B.S.; writing—review and editing, M.S., A., B.S., A.R., S.K.T., V.M., S.K. and D.S.; supervision, B.S.; funding acquisition, B.S. All authors have read and agreed to the published version of the manuscript.

**Funding:** This research was funded by Haryana State Council for Science and Technology, Panchkula, Haryana.

**Institutional Review Board Statement:** Not Applicable.

**Informed Consent Statement:** Not Applicable.

**Data Availability Statement:** The data presented in this study are available on request from the first and corresponding authors.

**Acknowledgments:** The authors acknowledge the Haryana State Council for Science and Technology, Panchkula, Haryana for the financial assistance in the form of a research project (HSCST/R&D/2017/62) and fellowship (HSCST No. 1743, dated 12/4/2017) during the tenure of this research work.

**Conflicts of Interest:** The authors declare no conflict of interest.

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
