# Peer review of "Bioacetoin Production by Bacillus subtilis subsp. subtilis Using Enzymatic Hydrolysate of Lignocellulosic Biomass"

_fermentation, doi:10.3390/fermentation9080698_

Round 1

Reviewer 1 Report (Previous Reviewer 4)

The manuscript has been improved significantly  after major revision. Hence, the revised manuscript can be accepted for publication.

Reviewer 2 Report (Previous Reviewer 5)

In this work, the authors studied the production of bioacetoin by a wild strain of B. subtilis subsp. subtilis JJBS250 under optimized culture conditions in submerged fermentation. Substrates used consisted in enzymatic hydrolysate of pretreated lignocellulosic biomass.

It is interesting and original, it can be published as it is, now. 

This manuscript is a resubmission of an earlier submission. The following is a list of the peer review reports and author responses from that submission.

Round 1

Reviewer 1 Report

The submitted manuscript (ID: Fermentation-2395265) aims to culture conditions and nutritional requirements optimized for bioacetoin production by wild strain of B. subtilis subsp. subtilis JJBS250, an eco-friendly and economical method for the production of bioacetoin. The manuscript is very interesting and provides new ideas for the biodegradation of lignocellulosic biomass. However, several issues need to clarify before the manuscript can be considered for possible publication.

1.      In the introduction part, the language is too cumbersome, and the logic is not strong. It is suggested to modify this section for clearer expression. In Part 2.7, there is the problem of repeatedly citing the same references, so it is suggested to review and revise. And the cited reference is too old, please consider the most recent reference such as Bioresource Technology 2023, 369, 128390; Catalysis Today, 2022, 404, 35-48.

2.      In the "Results and Discussion" section on pages 4 to 5, a lot of background is written, suggesting changes to the content of this section as well as the language logic.

3.      Can you explain why the concentration of the peptide in Figure 2 (b) shows a marked drop at 1.5% and then a rise at 2%?

4.      The missing punctuation marks and incorrect unit writing in the article are shown as "g/1" in Table 1. It is suggested to review other places and correct them.

In the introduction part, the language is too cumbersome, and the logic is not strong. It is suggested to modify this section for clearer expression. 

Reviewer 2 Report

The average bacterial acetoin production was around 50-100g/L, whereas this study only produced a maximum titer of 1017 mg/L. The research design is very simple and nothing important is revealed. It lacks interests to readers.

Reviewer 3 Report

Dear Authors, 

Thank you for submitting the manuscript to Fermentation! The manuscript though provides a good topic, but the methodology used does not stands fare with the latest advancement in the field of bioprocessing. There are several papers available on the advancement in bioprocess optimization. OFAT methodology use as a stand alone is an obsolete practice. The OFAT is generally well supported with DOE or a procedure where the results from OFAT have been tested in scale up and showed the robustness and reliability of the bioprocess. Even DOE methodology has progressed toward using neural link optimization to further improve the process of optimization. The use of statistics should be from the beginning of the study i.e. experimental design. Using statistics on the results obtained does increases the confidence in the result by reducing standard error, but it cannot remove the variability due to the biasness in the experimental design. 

In the methods for the source of bacterial strain the information on isolation is in other manuscript. This hampers the flow for the readers as they are forced to spend time into other places to understand the methodology. The readers should be able to get all the information in one place. 

The figures do not show the lower level of the standard errors. The corresponding author should ensure that the figure represented are  in the correct format. 

Once, the authors found that the organic nitrogen are good for producing bioacetoin and it is due to availability of the other elements viz. minerals, vitamins etc. why the authors did not try to find if the increase is due to organic nitrogen or due to other elements. 

The figure 5 showing TLC, there is no standard. How did the authors knew that this is the compound that is being investigated?

The authors never discussed what caused the increase in production with addition of peptone? Was it due to increase in the microbial inoculum or due to ability to maintain the microbial growth in the product production. The authors should have supported this with data to know at which phase of the growth the product is being product. A good paper to refer on this to understand all the points mentioned is https://www.mdpi.com/2076-2607/6/3/93.

The results have been discussed and supported by the references, but why the results were obtained as such is missing in the discussion. forexample: it was neve mentioned why sucrose produced higher product that glucose which the authors explained that it is the best carbon source due to its ease of assimilation and metabolism by bacterial culture. 

Thanks

None

Reviewer 4 Report

Dear Authors,

The research study performed in the current MS is overall  good. But still some sections need to be improved. I have given my comments below.  

Title: Bioacetoin production by Bacillus subtilis subsp. subtilis using  enzymatic hydrolysate of lignocellulosic biomass

  • Comments
  1. In this study, authors have used wild type subtilis subsp. subtilis JJBS250 as a potent bacterial strain for utilization of renewable and cost-effective lignocellulosic hydrolysates for economical bioacetoin production. Enzymatic hydrolysates supplemented with peptone showed 2 to 4-fold improvement in the production of bioacetoin production
  2. The research topic is interesting and article is well written considering explanation of the subject, however there are some flaws which needs to be improved.
  3. As shown in Fig. 3, maximum acetoin production was obtained using sucrose and not in simple sugar glucose? Please explain it.
  4. There are several articles have been published on the topic related acetoin synthesis using different approaches. So Please highlight that the uniqueness of your studies.
  5. As compared to other reported studies, the acetoin production levels are very low. So Include one table as comparative explanation of the processes reported till date by other researchers.  
  6. Please improve the font sizes in all figures
  7. I suggest to add some relevant and significant references related to enzymatic degradation process e.g. (1) Lignin valorization using biological approach.Biotechnology and Applied Biochemistry 68 (3), 459-468

    (2) Current developments in lignocellulosic biomass conversion into biofuels using      nanobiotechology approach Energies 13 (20), 5300

  1. According to my opinion, this article can not be accepted in the current state.

Reviewer 5 Report

In this work, the authors studied the production of bioacetoin by a wild strain of B. subtilis subsp. subtilis JJBS250 under optimized culture conditions in submerged fermentation. Substrates used consisted in enzymatic hydrolysate of pretreated lignocellulosic biomass. 

It is interesting and original. Before the publication, some aspects need minor insights, such as: 

1. Abstract: The used substrate (hydrolysate?) is not clear. Better a short description of the experimental activities with respect on a lot of confused results. It has to be improved, rewrite it!

2. Introduction: 2nd generation sugars (obtained by hydrolysis of lignocellulosic biomass) can be used to produce a wide range of bioproducts. Cite some examples e.g. Caporusso, A., De Bari, I., Giuliano, A., Liuzzi, F., Albergo, R., Pietrafesa, R., Siesto, G., Romanelli, A., Braccio, G., Capece, A., 2023. Optimization of Wheat Straw Conversion into Microbial Lipids by Lipomyces tetrasporus DSM 70314 from Bench to Pilot Scale. Fermentation 9, 180. https://doi.org/10.3390/fermentation9020180;

3. Introduction: the industrial application of this study isn’t clear. Authors have to deepen the potential industrial applications of their research. First with a specific bibliography analysis about the industrial needs of the research section of the work, then describing the final purpose of the work done, in terms of industrial application;

4. Discussion: describe the results obtained and how the current enzymatic hydrolysate fermentation processes can improve.

5. Discussion and conclusions: the research can be interesting, but the application of this study isn’t clear. Authors have to deepen the potential industrial applications of their research, also extending the same analysis to other countries;

6. In general, the authors should describe the advantages of the adopted approach and obtained results also considering the potential environmental/economic improvement.

Some minor English spellings have to be resolved.